# Piercing the corporate veil system and creditors protection: Based on the investigation of debt paying ability—Empirical evidence from the 2005 *Company Law* amendment

**Jun Tian**☯, **Zuopeng Chen**✆*☯, **Yue Zhu**

School of Economics and Business Administration, Heilongjiang University, Harbin, Heilongjiang Province, China

☯ These authors contributed equally to this work.

\* a15561818552@163.com

**Data Availability Statement:** Data are available from the Wind Information(https://www.wind.com.cn/portal/zh/EDB/index.html). Because the data is

## Abstract

This paper uses the difference-in-differences model to research how the "piercing the corporate veil" system marked by the 2005 *Company Law* amendment affects the level of corporate creditor protection. The research results show that private enterprises and local state-owned enterprises are sensitive and significant to this legal amendment. In contrast, local state-owned enterprises are more sensitive and have a stronger motivation to protect the interests of creditors. The motivation of companies with weaker profitability for creditor protection lasts not only for the year of law revision but also extends to the year of implementation. With the law's implementation, the growth effect of creditor protection for local state-owned enterprises has become more significant. Further analysis shows that the main findings of this article are more significant in companies with larger debt scales, companies with a higher year-on-year growth rate of operating income, companies with controlling shareholders, and companies with higher stock market capitalization. From an empirical research view, this paper explains the economic effect and mechanism of the whole corporate personality under the complete system and adds economic evidence for how the law acts on the capital market.

## Introduction

Since the promulgation of the *Company Law* in 1993, China's capital market has developed rapidly in the globalization trend of the world economy, providing top-level protection for the orderly operation of the capital market. The old *Company Law*, which strictly limited the autonomous management rights of companies, could no longer meet the development needs of the integrated world economy, and an adequate, top-down legislative reform was urgently needed to meet the needs of China's corporate development. Against this background, the

third-party, the author has no right to publish, other researchers can enter the website according to the guide registration to obtain data.

**Funding:** This study was supported by the following sources of funding: "Research on mechanism and countermeasures of regional employment selection of college graduates from the perspective of highly educated population loss in Heilongjiang Province" from Heilongjiang Provincial Office of Philosophy and Social Sciences awarded to JT. The funders had no role in study design, data collection and analysis, decision to publish, or preparation of the manuscript.

**Competing interests:** The authors have declared that no competing interests exist.

2005 revision of the *Company Law* is a far-reaching legislative reform compatible with China's economic transformation.

At its inception, China's *Company Law* adhered to a strict legal capital regime, restricting the autonomy of shareholders. At the same time, government administrative organs abused their power. They interfered in the operation of the capital market, resulting in many short-comings in the judicial practice of applying the *Company Law*. This 2005 legislative amendment began to advocate the transformation of capital credits to asset credits while improving the corporate governance system. In addition to preventing the abuse of shareholders' rights and protecting the interests of creditors, the amendment historically provided for the system of "piercing the corporate veil", with the clear legislative intent of holding shareholders or contributors responsible for abusing the corporate personality of the company behind the scenes, and, in particular, the state-owned enterprises with abusive practices have become the key targets of the amendment.

Implementing the system of "piercing the corporate veil" has great significance,because the company is often referred to as a "contract connection". Basically, as the counterpart of many contracts with suppliers, employees, and customers, the company coordinates the behavior of many subjects through the exercise of contractual rights [1,2]. A company is a collection of individuals, and treating a company as a legal person is a legal fiction introduced to facilitate the company's business in a privileged manner [3]. Although the principle of limited liability of shareholders and the principle of independence of corporate personality have promoted the civilization and progress of modern material society, they have inevitably provided opportunities for shareholders to take advantage of the independence of corporate personality for their improper interests [4]. The system of "piercing the corporate veil" maintains the essence of establishing the legal person system and is the exception of the company's legal person.

The Anglo-American law system has a wide range of judicial applications of the system of "piercing the corporate veil". In the case of the United States v. Milwaukee Refrigerator Transportation Company in 1905, Judge Sanborn first created the system of "piercing the corporate veil" and finally became an important theory in *American company law* [5]. Justice Cardozo believes that the issue of "piercing the corporate veil" is "wrapped in the fog of metaphor", resulting in very different court decisions [6]. Since the rules of "piercing the corporate veil" may interact with other legal rules, there is still substantial confusion when listing many representative facts that constitute personality confusion [7]. Therefore, "piercing the corporate veil" is still a very dynamic principle in an era when neither the rules nor the capital system is immutable.

The application of the negative system of corporate personality in the continental law system is mostly derived from the provisions of *German Company Law*. The German perspective theory is the main legal basis for regulating the relationship between closed companies and creditors [8]. The German J*oint*-stock *Company Law* stipulates the chapter of affiliated enterprises, which stipulates the circumstances under which the parent company should be liable for the debts of its subsidiaries, which is equivalent to the role played by the principle of "piercing the corporate veil". It stipulates that the use of its influence on the company leads to damage to the company, and the person (including shareholders) who inflicts the damage shall be liable for damages to the company, the shareholders of the company, and the creditors of the company [9].

At present, the jurisprudence basis of the Anglo-American law system and continental law system in the field of creditor protection and the protection measures related to company law, try to prove the internal logic of creditor protection level from the theoretical level and construct a relatively complete theoretical framework to protect the interests of creditors [10–12]. However, in the field of economics, the research on the relationship between the

implementation of the system of "piercing the corporate veil" and the protection of creditors has not been extended to empirical research.

On October 27, 2005, several amendments including the "piercing the corporate veil" system were passed at the 18th meeting of the Standing Committee of the 18th National People's Congress of the People's Republic of China. The revision of the *Company Law* in 2005 is an important legal reform event, that is of great significance in the history of Company *Law* reform. This article regards this legal amendment as a rare quasi-natural experiment to analyze the impact of the amendment of *Company law* on creditor protection, trying to reveal the impact of this legal amendment on the protection of the interests of creditors of the mechanism. We use the difference-in-differences model (DID) to study how the "piercing the corporate veil" system marked by the revision of the *Company Law* in 2005 affects the protection level of corporate creditors.

Our study contributes to the literature in three ways:(1) From the perspective of economic logic, the level of creditor protection consists of two parts: solvency and willingness to repay debt. This paper argues that the *Company Law* follows the basic logic from improving solvency to improving willingness to repay debt in the redistribution of interests between equity and creditor's rights. Therefore, this paper follows this logic and innovatively constructs the creditor protection index, which enriches the application of the "Law and Finance" (LLSV) theory of creditor protection in China and fills the research gap on how to measure the level of creditor protection at the company aspect. (2) Using the 2005 *Company Law* amendment as a quasi-natural experiment shock at the legislative level, DID is used to empirically test the impact of the two quasi-natural experiment shocks on the interests of creditors. From the perspective of empirical tests, it is concluded that the corporate personality denial system has a positive effect on the interests of creditors, which adds an econometric empirical basis for the legal profession on how the two revisions of the *Company Law* affect the interests of creditors. (3) This paper innovatively tests the dynamic effect and actual effect of the revision of *Company Law* to improve the protection level of creditors of small and medium-sized enterprises. Companies of different natures show differences in dynamic promotion effect.

The remainder of the paper is organized as follows. The section Theoretical analysis and research hypothesis briefly reviews the literature and presents our hypothesis. The section Model design and research methodology explains the data and research design. The section Empirical results presents our empirical results. The section Conclusions and implications follows conclusions and implications.

## Theoretical analysis and research hypothesis

### The basis efficiency of "piercing the corporate veil" in the perspective of debt covenants

We first discuss the basic efficiency of "piercing the corporate veil" from the perspective of debt covenants. Debt contract theory is divided into equity-financed governance and debt-financed governance based on differences in governance. The so-called contractual governance effect mainly refers to the creditor signing a debt contract with the enterprise to ensure the safety of the contract and its interests and use the rights granted by the debt contract to supervise, control, and motivate the controlling shareholders of the enterprise [13].

Therefore, the freedom principle of contract is the common basis of agreement, signing, execution, and breach of contract. The law forces both parties to implement the responsibility of the contract agreement. Only in this way can the creditor-debtor relationship maintain the initial agreement in the contract. The system of "piercing the corporate veil" has played a role

in protecting the interests of creditors from the damage of controlling shareholders in the default stage of the debt contract [14].

Limited liability transfers shareholders' assets to the company's assets, converts shareholders' unlimited liability for the company's debts into limited liability, and facilitates the separation of the company's property from the shareholders' property. In the absence of a mandatory legal restriction, as the upper limit of limited liability capacity, contractual limited liability is no different from unlimited liability, and the upper limit of the maximum liability capacity of a contributor consists of the maximum amount of its assets. This means that, in the connection between shareholders and creditors through the debt contract, the creditors will consider whether the debtor company can repay the debt at the beginning of the transaction; at the same time, they will consider whether they have the property basis to complete the contractual transaction. Suppose the information between the two parties is entirely symmetrical,in that case, the creditor, to maximize the expected return, decides the upper limit of its liability capacity based on the size of the wealth of the company and the controlling shareholders and agrees on a contractual price that is commensurate with the return and the risk.

In the actual transaction, the information of both parties to the contract is incomplete. The creditors cannot observe the accurate scale of the company's capital when choosing the object of cooperation, especially under the legal capital system. The registered capital often becomes the "ruins" on the original balance sheet, which is unable to convey the debtor's natural ability to repay the debt, and because of this reason, the *Company Law* stipulates the principle of information disclosure and the principle of capital maintenance to match the limited liability system. Under market competition pressure, controlling shareholders may have incentives to abuse limited liability [15]. The specific performance is that shareholders with control in debt transactions have information advantages. If the "misleading" signal that the solvency is higher than the actual situation of the company is conveyed to the creditor through the "inflated capital" scale, the contract price with the nature of "fraud" is thus formed, and the controlling shareholder obtains excess personal income. At this time, the controlling shareholders deviate from the limited liability contract, resulting in damage to the interests of creditors.

When there is a causal relationship between the use of the controlling shareholders' rights and the abuse of limited liability, the law constructs a system of "piercing the corporate veil", i.e., the law ignores the independence of the personality of the corporate person. The "piercing the corporate veil" directly traces the actual situation behind the legal characteristics of the company, orders the relevant shareholders to directly assume the obligations and responsibilities of the company and corrects the possibility that the control right deviates from the initial contract, so that it bears unlimited liability to the creditors whose interests are damaged. As a result, the scope of debt repayment for creditors is expanded from the company's asset cap to the maximum asset amount of controlling shareholders, and the credit base is re-transferred from the company's assets to the shareholders' assets, which ultimately improves the company's ability to repay debts [16].

## The "piercing of the corporate veil" system in the perspective of the "hawk-dove" game

In judicial practice, the system of "piercing the corporate veil" is a relief measure at the stage of corporate litigation, an essential means of correcting the deviation of controlling shareholders from the initial contract in the *Company Law*, and a necessary system for resolving disputes over breach of contract in the contract design. The "Hawk and Dove Game" helps to explain the role of "piercing the corporate veil" to improve the solvency of the debtor company, effectively compensating for the interests of creditors. This is shown in **Fig 1**. In this game, creditors

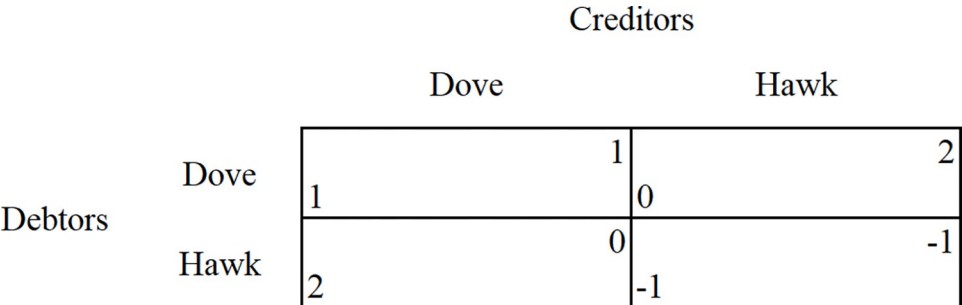

**Fig 1. The hawk and Dove Game.**

and debtors choose between an aggressive "hawk" strategy and a passive "dove" strategy. The "hawk" and the "dove" are two different strategies, i.e., the aggressive and the submissive strategies, which can also be interpreted as the bent and the withdrawal strategies. In the debt transaction, four kinds of game results are obtained: (1) hawk vs dove, (2) dove vs dove, (3) dove vs hawk, (4) hawk vs hawk, and the pure strategic equilibrium is hawk vs. dove and dove vs. hawk, that is, both parties in the contractual transaction adopt the strategy of co-operation, and the hawk vs. hawk is the costliest co-operation strategy for both parties to the contract [17].

When the law does not explicitly provide for a system of "piercing the corporate veil", controlling shareholders abuse the limited liability system by inflating the size of capital to send false signals to creditors about their ability to be held liable because the corporate legal system constitutes a legal barrier between creditors and shareholders of the company. Creditors can only pursue the liability of the debtor company by adopting a "confrontational" strategy rather than directly pursuing the liability of the infringing shareholders. As the eagle vs. eagle confrontation strategy was already in place, there was already a false increase in the size of the company's capital at the time of the establishment of the deed. At this time, the company's debtor's solvency could not make up for the loss of creditors' interests, which leaded to the maximization of the loss of creditors' and debt companies' interests, while the infringing shareholders obtained excess private benefits.

The principle of legal protection of contractual interests is tilted in favor of the interests of third-party creditors. The introduction of the system of "piercing the corporate veil" shifts the credit base of the company, shifting corporate assets to shareholders' assets, and, for (external) creditors whose interests have been harmed, widening the scope of debt repayment (the solvency of the debtor company) to make up for the loss of creditor interest. When the losses of controlling shareholders rise, they are forced to return to the dove strategy, i.e., the coercive role of the law in resolving default disputes, and give in to the hawk strategy of creditors (Adams, 2000).

From the above analysis, it can be concluded that the design of this system is such that the law serves to increase the controlling shareholders' willingness to pay their debts at the final stage of the contract, i.e., after default has occurred, and not at the initial stage of the contract's formation. While this system can force the expansion of the scope of creditor compensation, it does not improve the incentives for debtors to repay their debts. Therefore, the law should design the system so that it not only improves the debtor's solvency from an ex-post perspective but also attempts to improve the debtor's willingness to pay from an exante perspective (e.g., authorized capital regimes).

It can be concluded that the system of "piercing the corporate veil" has the function of changing the credit basis of the company and improving the debtor's Solvency. In the complex

problem of preventing the abuse of limited liability by the controlling shareholders, it corrects the possibility of deviation of the control from the initial contract by piercing the "veil" of the company. It plays the coercive role of the law by imposing the responsibility of repaying the debt to the controlling shareholders law's coercive role.

In addition, the law's guiding, evaluating, predicting, and educating role makes this amendment "indirectly" increase the willingness of controlling shareholders to repay as external pressure. Specifically, the implementation of the "piercing the corporate veil" system in the *Company Law* can "guide" the controlling shareholder to return to the fair "consideration" of the contract and conduct equal transactions based on meeting their solvency. Once there is a "willingness" to deviate the control right from the initial contract, shareholders can predict that such behavior will be sanctioned by law, but improving the willingness to pay debts through guidance and prediction under external pressure is not the main form of "piercing the corporate veil" system. Therefore, we believe that the 2005 *Company Law* amendment was mainly to strengthen the creditor's rights constraint by improving the debtor's solvency.

## Background and content of the implementation of the "piercing the corporate veil" regime

The system of "piercing the corporate veil" in major countries of the world consists of the following elements: (1) The corporate independence of the company is ignored. There is a mixing of shareholders and corporate property. (2) The company is grossly undercapitalized. There may be situations where the controller of the company establishes the company for his benefit. (3) Concealment or distortion of the proper operating information of the company, transactions with misrepresentation to creditors, or other fraudulent behavior [18–20].

Before the revision of the *Company Law* in 2005, there were a large number of local state-owned enterprises with concentrated equity, and the events of major shareholders' tunneling of listed companies continued to occur. The abuse of the independent personality of the company's legal person and the limited liability of shareholders by the company's owners has been repeatedly prohibited in China's limited liability companies and joint stock limited companies, which has seriously damaged the interests of small and medium shareholders and creditors. Some companies try to use corporate personality and shareholder limited liability to evade tax obligations, infringe on the interests of employees, and create involuntary creditors in the company's operations or difficulties.

Therefore, in the context of the characteristic socialist market economy of China's economic transformation, the system of "piercing the corporate veil" was established in the revision of China's *Company Law* in 2005, which mainly contains the following elements:(1) The subject of the act must be the company's shareholders and cannot be the managers and other executives. (2) The victim is the creditor, not the other shareholders or other persons. (3) The consequence of the act is that the shareholders are jointly and severally liable for the company's debts. (4) The perpetrator has evaded the debt. (5) There is a causal relationship between the evasion and the shareholder's abuse of the company's independent legal person status and limited liability of shareholders. (6) The evasion of the debt has resulted in severe damage to the interests of the company's creditors.

However, as centralized state-owned enterprises are spread across the lifeblood industries of the national economy, with vital fixed assets and general profitability, their budgetary soft constraints are more likely to be more significant. In their dealings with adjustable creditors (mainly referring to state-owned banks), the property rights of such creditors and centralized state-owned enterprises are homogeneous, which determines that the implementation of the

system of "piercing the corporate veil" has no substantial impact on the level of creditor protection of centralized state-owned enterprises.

Before the amendment of the *Company Law* in 2005, a large number of local state-owned enterprises with concentrated shareholdings existed in China, incidents of major shareholders hollowing out listed companies were constantly occurring, and the company's fixed assets were limited, and the scale of assets is much smaller than that of the central state-owned enterprises. Therefore, the implementation of the "piercing the corporate veil" system is of great significance for reducing the abuse of corporate legal personality by local state-owned enterprises and promoting the improvement of the protection level of creditors of local state-owned enterprises with poor profitability.

Private enterprises have the weakest assets, the scale of enterprise capital is generally smaller than that of state-owned enterprises, fixed assets for collateralized lending are limited, budgetary constraints are strong, and enterprises are susceptible to legal requirements, so the implementation of a system of "piercing the corporate veil" will significantly improve the solvency of private enterprises and increase the level of protection for creditors.

In addition, in the stage of amending the *Company Law* in 2005, private enterprises, local state-owned enterprises, and central state-owned enterprises are facing the same legal regulation background, subject to the same information disclosure system regulation and the same market competition environment. Under external pressure, the three types of enterprises have the same willingness to repay debts. In the context of this legal amendment, the willingness to repay debts is affected by external pressure and does not have the motivation to improve internally. Based on the above analysis, this paper puts forward the research hypothesis:

**Hypothesis:** The 2005 *Company Law* amendment significantly improved the level of creditor protection of small and medium-sized listed companies.

## Model design and research methodology

### Creditor protection index construction

Referring to the investor protection index constructed by La Porta et al. (1998) [21] and the anti-self-dealing index created by Djankov et al. (2003) [22], this paper uses solvency and willingness to pay to construct explained variables of the level of creditor protection (*protect*) through RAGA-PPC model. Specifically, according to the availability of data, after summarizing the rationality and limitations of the existing literature to measure the level of creditor protection, the use of long-term and short-term solvency indicators relatively objectively and truly reflects the ability of debt companies to repay their creditors, and from corporate behavior events to prove whether the enterprise has the willingness to repay.

Based on the advantage that the economic dimension reduction model is easy to eliminates the weakness of the useful information of the indicators, the selected indicators (**Table 1**) are reduced in dimension, and the representative indicators selected by the RAGA-PPC model are reduced to one-dimensional space to construct the level of debt enterprises to protect the interests of creditors (*protect*). We use the RAGA-PPC model to evaluate the *project*, because it overcomes the weaknesses in the dimensionality reduction of principal component analysis and factor analysis, and is suitable for the dimensionality reduction measurement of the level of creditor protection. However, this indicator does not directly equate debt servicing capacity to the level of creditor rights protection. We choose listed firms in imperfectly competitive markets, which inevitably need legal norms to safeguard and regulate, but this does not mean that all firms will abide by contractual agreements under the constraints of the law, and there may exist a small portion of firms that default on debt repayment even when their solvency permits and other complex soft constraint situations.

**Table 1. Selection of indicators for creditor protection systems.**

| First level indicators | Secondary level indicators | Description of secondary level indicators | Indicator use |
|---|---|---|---|
| Short-term solvency | Current ratio | Current assets / Current liabilities | Evaluation of overall liquidity of current assets |
| | Quick ratio | (Current assets—net inventories) / Current liabilities | Solvency reflecting the realization of quick-liquid assets |
| | Cash-to-maturity debt ratio | Net cash flows from economic activities / (Short-term borrowings + Non-current liabilities due within one year + Notes payable) *100% | Reflects the ability of the enterprise's operating cash flows to repay current liabilities as they fall due |
| Long-term solvency | Cash flow interest coverage multiples | Net cash flows from operating activities / Interest expense | Reflects how many times operating cash flow per unit of interest expense is guaranteed as repayment |
| | Equity ratio | Total liabilities/total shareholders' equity | Reflects the capital structure of the company |
| | Total cash-debt ratio | Net cash flows from operating activities/Total liabilities | Reflects the ability of the enterprise's operating cash flows to service its overall liabilities |
| | Interest earned multiple | EBIT/Interest expense | Reflects multiple of EBITDA per unit of interest cost as solvency margins |
| Corporate Behavior Events | Whether to disclose major matters of the company promptly | "Yes" is 0.25 points, and "no" is 0 points. | This reflects whether the company does not disclose major matters promptly. |
| | Whether there is a false information disclosure | "Yes" is 0.25 points, and "no" is 0 points. | This reflects the impact of the reliability of information disclosure on creditors' decision-making. |
| | Are periodic reports disclosed promptly? | "Yes" is 0.25 points, and "no" is 0 points. | This reflects the impact of regular disclosure of company reports on creditors' decision-making. |
| | Are there company business warnings with "uncertain "results? | "Yes" is 0.25 points, and "no" is 0 points. | This reflects whether there is a risky business in the company's operation. |
| | Are there related party transactions? | "Yes" is 0.25 points, and "no" is 0 points. | This reflects the impact of the company's related party transactions on the overall interests of creditors. |
| | Whether there is a joint liability guarantee? | "Yes" is 0.25 points, and "no" is 0 points. | This reflects the impact of joint and several liability guarantees on the overall interests of corporate creditors. |

Note: To compare the three types of enterprise-scale more intuitively, this paper does not log the "company size" in the descriptive statistics but does descriptive statistics on the "company size" in the unit of 100 million RMB.

So we added an indicator of willingness to pay to address this shortfall. Based on the above analysis, we refer to the investor protection index constructed by La Porta et al. (1998) [21] and the anti-self-dealing index created by Djankov et al. (2008) [23], based on measuring the company's solvency, the corporate behavior events affecting creditor protection are added to prove whether the company has the willingness to pay the debt. The addition of corporate behavior events is more appropriate in this discussion for the specific following reasons. In the creditor's rights and debt relations, the debtor is not only required to have the ability to repay the debt but also restricted by the subsequent corporate behavior events. Because the market in which the research object is located is imperfectly competitive, legal norms are inevitably needed to guarantee and regulate. Therefore, it cannot be assumed that the attitude of legal norms to all companies to repay their debts is rigid. Constraints also mean that there may be a small number of companies that still default and do not repay their debts under the conditions that their solvency allows. Therefore, it is necessary to use the corporate behavior events that affect the interests of creditors as binding indicators to prove whether the company has the willingness to repay debts through the company's actual business behavior. Specifically, this

paper refers to the indicators selected by the existing research, mainly from the perspective of behavioral events that affect the interests of creditors. The construction of creditor protection indicators is shown in **Table 1**.

## Data samples and variables

In this paper, private enterprises, local state-owned enterprises, and central state-owned enterprises in China's Shenzhen Component Index (SSEC) and Shanghai Component Index (SZI) from 2000 to 2010 are selected as research subjects. Financial enterprises, ST enterprises, and enterprises delisted after 2005 are excluded. Due to the different listing years of enterprises, unbalanced panel data are obtained. To control for other firm factors that may affect the level of creditor protection, this paper introduces firm size (*Size*), financial leverage (*Leverage*), debt size (*Debt*), equity multiplier (*Equity*), increase rate of main business revenue (*Receipt*), return on equity (*ROE*), fixed asset ratio (*Fixed)*, and the proportion of independent directors (*Director*). Specifically, *Size* is defined as the logarithmic value of *Total Assets*. *Leverage* is defined as *EBIT/(EBIT− Finance Costs)*. *Debt* is defined as *Total Liabilities/Total Assets*. *Equity* is defined as *Total Assets/Total Shareholders' Equity*. *Receipt* is defined as (*Current Operating Income − Last Period Operating Income*)/ *Last Period Operating Income*. *ROE* is defined as *Net Profit/ Net Assets*. *Director* is defined as the *Number of Independent Directors/Number of Board of Directors*.

The descriptive statistics of the corporate operating indicators of private enterprises, local state-owned enterprises, and central state-owned enterprises involved in the paper are shown in **Table 2**.

## Model setting

When using quasi-natural experimental methods to study how systems such as "piercing the corporate veil" affect the level of creditor protection, it is necessary to exclude the impact of factors other than legal amendments on the level of creditor protection through group comparison. Based on the above reasons, this paper sets the grouping standard as the company's profitability and specifically uses the return on assets (*ROA*) as the grouping standard. This paper draws on the treatment methods of Vig (2013) [24] and Campello (2016) [25]. Specifically, the control group and the experimental group are constructed according to the return on assets of listed companies. The reasons for this treatment are as follows:

Firstly, when a company is in the process of operation or when the company is insolvent, it shows a significant difference in the dimension of the characteristic of profitability. The company's profitability is in a dynamic process of change, the stronger the company's profitability in the current year, indicating that the company in the process of operation, the more profits, and thus has more retained earnings, which in turn affects the next year's financial budget and financing decisions. Specifically, by increasing the portion of financing retained internally, the proportion and risk of external financing can be reduced accordingly, which is conducive to improving the ability to repay debt to existing and future creditors.

Secondly, the revision of the *Company Law* in 2005 provides for the "piercing of the corporate veil" system, directly targeting the interests of creditors to put forward a more powerful means of protection, that is, the economic subject can negate the independent personality of the company directly to the responsible shareholders, the interests of the main body affected by this principle is bound to be the profitability of the enterprise is relatively poor. This paper argues that enterprises with a lower ability to create wealth or make profits are less capable of repaying their debts to creditors, and after the law is amended, enterprises will inevitably improve their debt repayment ability through increasing internal retained earnings or

**Table 2. Descriptive statistics of variables.**

| Type of enterprise | Variables | Observations | Mean | Standard deviation | 25 percent quartile | 75 percent quartile |
|---|---|---|---|---|---|---|
| Private enterprises | Size | 2816 | 17.322 | 21.633 | 5.465 | 20.400 |
| | Leverage | 2832 | 1.529 | 9.262 | 0.511 | 1.653 |
| | Debt | 2393 | 1.038 | 9.674 | 1.593 | 2.693 |
| | equity | 2701 | 3.006 | 9.674 | 1.593 | 2.693 |
| | Fixed | 2446 | 0.259 | 0.161 | 0.137 | 0.361 |
| | Receipt | 2743 | 31.862 | 129.882 | -3.663 | 35.235 |
| | ROE | 2697 | 3.494 | 35.670 | 2.723 | 13.221 |
| | Stock | 2446 | 0.259 | 0.161 | 0.137 | 0.361 |
| | Director | 2726 | 0.294 | 0.137 | 0.273 | 0.364 |
| Local state-owned enterprises | Size | 5893 | 35.466 | 51.168 | 9.721 | 36.411 |
| | Leverage | 5892 | 1.493 | 9.872 | 0.535 | 1.670 |
| | Debt | 5892 | 0.432 | 0.257 | 0.279 | 0.575 |
| | equity | 5841 | 1.665 | 10.344 | 1.560 | 2.689 |
| | Fixed | 5942 | 0.327 | 0.194 | 0.179 | 0.461 |
| | Receipt | 5823 | 19.129 | 37.786 | 0.346 | 30.782 |
| | ROE | 5837 | 5.016 | 15.219 | 2.463 | 10.750 |
| | Stock | 6117 | 21.742 | 0.987 | 21.072 | 22.306 |
| | Director | 5804 | 0.281 | 0.136 | 0.222 | 0.364 |
| Central state-owned enterprises | Size | 2653 | 598.768 | 6564.136 | 9.426 | 53.317 |
| | Leverage | 2310 | 1.370 | 6.284 | 0.489 | 1.645 |
| | Debt | 2549 | 0.455 | 0.597 | 0.245 | 0.574 |
| | equity | 2309 | 2.616 | 4.460 | 1.537 | 2.665 |
| | Fixed | 2065 | 0.292 | 0.194 | 0.150 | 0.406 |
| | Receipt | 2299 | 24.790 | 76.927 | 1.500 | 32.574 |
| | ROE | 2304 | 9.366 | 153.954 | 2.723 | 12.635 |
| | Stock | 2414 | 147.835 | 983.930 | 21.320 | 37.261 |
| | Director | 2263 | 0.286 | 0.150 | 0.250 | 0.333 |

expanding external financing, effectively increasing the level of creditor protection of enterprises, Therefore, it is more motivated by the revision of the law. Enterprises with a greater capacity for wealth creation, on the other hand, will have a greater capacity to repay their creditors and will be less affected by changes in the law, regardless of whether the law is changed or not, in other words, whether the system of "piercing the corporate veil" is implemented or not.

Based on the above analysis, the impact of the 2005 *Company Law* amendment on the level of creditor protection varies among companies with different profitability levels: for companies with low profitability levels, the motivation to protect the interests of creditors is stronger; for companies with higher profitability levels, their sensitivity to the law change is relatively weak, and their asset allocation and financial plans are less likely to be affected by external policy shocks and have limited impact on the level of protection of creditors' interests.

Specifically, this paper calculates the average value of the return on assets of the sample enterprises in the two years before 2005 (i.e.2003 and 2004) and uses the 33% and 67% quantiles of the average value as the threshold value. According to the return on assets, the samples are divided into three groups: the highest 1 / 3, the middle 1 / 3, and the lowest 1 / 3. On this basis, the 1 / 3 with the highest return on assets is defined as the control group, and the 1 / 3 with the lowest return on assets is defined as the experimental group.

In this paper, to eliminate the differences between individuals and time, the DID with two-way fixed effects is used to conduct empirical tests in three types of enterprises- private listed

enterprises, local state-owned listed enterprises, and central state-owned listed enterprises, respectively, and the design of model (1) is shown below:

$$Protect_{it+1} = \alpha + \beta_1 Low_i \times Year2005_t + \beta_2 Low_i + \beta_3 Year2005_t + \beta_4 Con_{it} + \delta_i + \gamma_t + \varepsilon_{it} \quad (1)$$

where $i$ in the model (1) denotes enterprises, $t$ denotes time, and the explanatory variable ($Protect_{it+1}$) is the level of creditor protection of firms measured by the RAGA-PPC model. According to the characteristics of profitability grouping and the research purpose of this paper, advancing the explanatory variables by one period is more in line with the purpose of this paper's research and further attenuates the unfavorable impact of endogeneity; the grouping indicator variable is $Low_i$, when the enterprise belongs to the experimental group, the enterprise's return on assets is in the lowest 1/3 group, the variable takes the value of 1; when the enterprise belongs to the control group, the enterprise's return on assets is in the highest 1/3 group, the variable takes the value of 0; $After_{it}$ is also an indicator variable, taking the value of 1 when the sample observation occurs in the year of the 2005 *Company Law* amendment and the subsequent years, and 0 otherwise; $Con_{it}$ is a control variable at the firm level; $\delta_i$ is an individual fixed effect, and $\gamma_t$ is a time fixed effect; $\varepsilon_{it}$ is the error term; and the statistical standard errors of the measured results have been adjusted for clustering at the firm level.

## Empirical results

### The time trend of the level of creditor's protection interests

This paper divides the A-share listed enterprises in SSEC and SZI into three categories: private enterprises, local state-owned enterprises, and central state-owned enterprises, and discusses the impact of the "piercing the corporate veil" system on creditor protection. By drawing the mean value of each group of enterprises, it can be seen that the overall trend of creditor protection level of the three types of enterprises is similar, but there are differences in local extreme points. In 2005, there was a local maximum point for private enterprises and local state-owned enterprises, while the central state-owned enterprises were not significant. It preliminarily shows that the average value of creditor protection level of the first two types of enterprises may be affected by the impact of legal changes under the original trend, while the central state-owned enterprises have a weak trend, which is consistent with the economic intuition that the company is large, free capital is sufficient, and the nature of the enterprise is special. From the trend of **Fig 2**, we can preliminarily get the inconsistency of the three types of enterprises under the law modification. The overall trend is similar to the economic intuition, and the detailed differences between the three types of enterprises will be obtained from the empirical test results of the DID below.

### The empirical test results of benchmark regression

To examine the impact of the 2005 *Company Law* amendment on the level of creditor protection, this paper uses model (1) to conduct a DID test on three types of enterprises: private listed enterprises, local state-owned listed enterprises, and central listed enterprises. The specific empirical test results are shown in **Table 3**.

From **Table 3**, it can be seen that the core explanatory variables of private enterprises and local state-owned enterprises are significantly positive, which are significant at the level of 10% and 1% respectively, while the core explanatory variables of central state-owned enterprises are not significant. At the same time, the core explanatory variable coefficient of local state-owned enterprises is slightly larger than that of private enterprises.

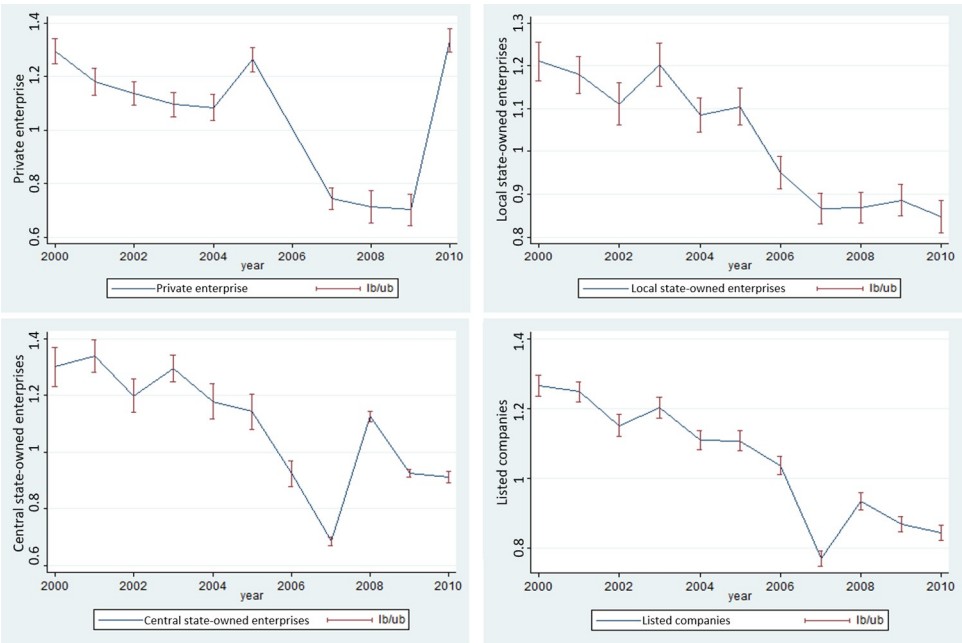

**Fig 2. Time sequence curve in the level of protection of creditors' interests from 2000 to 2010.**

We believe that this is related to the following two reasons. First of all, the *Company Law* in the 2005 amendment established the system of "piercing the corporate veil", clarified the loyalty and diligence obligations of directors, and stipulated the independent director system, which greatly reduced the possibility of controlling shareholders abusing limited liability. Compared with central state-owned enterprises, private enterprises and local state-owned enterprises have smaller scale and higher financial risk. Compared with similar enterprises with higher profitability, their business reputation and financing channels are more vulnerable. In the reform of the "piercing the corporate veil" system, enterprises with weak profitability have increased market competition pressure, urgently need to improve their debt repayment ability, and have more urgent motivation to protect the interests of creditors, which has promoted small and medium-sized listed enterprises such as private and local state-owned to improve their solvency.

Moreover, local state-owned enterprises are more significantly affected by this policy, indicating that the 2005 *Company Law* amendment has touched the interests of controlling shareholders in local state-owned enterprises. Compared with private enterprises, the financial management decision-making of local state-owned enterprises is more sensitive to legal changes, which enhances their ability to repay creditors.

Finally, the insignificant results of the 2005 *Company Law* amendment on the central state-owned enterprises show that the central state-owned enterprises are less affected by external policies due to their strong capital scale, small asset-liability ratio, diversified access to loan support, and easy access to mortgage loans.

## The empirical test results of the dynamic effect

To further study the mechanism of the 2005 *Company Law* amendment, this paper verifies the dynamic changes in the level of creditor protection before and after the 2005 *Company Law* amendment. First of all, if the level of creditor protection before the 2005 *Company Law*

**Table 3. The empirical test results of benchmark regression.**

| Protect | Private enterprises | Local state-owned enterprises | Central state-owned enterprises |
|---|---|---|---|
| $Low_i \times Year2005_t$ | 0.098* | 0.137*** | 0.062 |
| | (0.057) | (0.025) | (0.042) |
| Size | 0.0001 | 0.0001 | 0.010 |
| | (0.0004) | (0.0001) | (0.020) |
| Fixed | -0.007 | 0.242 | -0.0003 |
| | (0.156) | (0.074) | (0.001) |
| Tobin | 0.002 | -0.012 | -0.006 |
| | (0.002) | (0.009) | (0.005) |
| Leverage | 2.211 | 0.006 | 0.002 |
| | (3.003) | (0.004) | (0.004) |
| Debt | -0.389*** | 0.003* | 0.002 |
| | (0.132) | (0.002) | (0.001) |
| ROE | -0.001** | 0.0003* | 0.002*** |
| | (0.001) | 0.0002 | (0.001) |
| equity | -2.209 | -0.003 | -0.017 |
| | (3.003) | (0.003) | (0.010) |
| Receipt | -0.0003 | -0.0001 | 0.0002* |
| | (0.001) | (0.0001) | (0.0001) |
| Stock | 0.004 | 0.029 | 0.00002 |
| | (0.015) | (0.017) | (0.00003) |
| Director | -0.520*** | -0.088 | -0.061 |
| | (0.079) | (0.078) | (0.079) |
| Constant | 3.626 | 0.502 | 0.601*** |
| | (3.003) | (0.358) | (0.033) |
| Firm_fe | YES | YES | YES |
| Time_fe | YES | YES | YES |
| Observations | 1642 | 3057 | 1644 |
| Rho | 0.356 | 0.305 | 0.334 |

Note: Standard errors in parentheses:* p<0.1、 ** p<0.05、 *** p<0.01. The same for the following tables.

amendment did not have a significant impact, and only began to have an impact in the year of amendment, then the result is considered to be robust. If the company has made a significant change in the level of creditor protection before the 2005 *Company Law* amendment, it will interfere with the causal relationship between the change of the law and the protection of creditors. Secondly, the continuous impact of the amendment of the law on the protection of creditors for a long time, not only in the year of amendment but also in the specific implementation year, that is January 1, 2006.

The impact of the revision of the *Company Law* in 2005 on the level of creditor protection is dynamically tested from the above perspectives. Referring to the model settings of Bertrand and Mullainthan (2003) [26], the design model is shown in (2).

$$
\begin{aligned}
Protect_{it+1} = {} & \alpha_0 + \delta_1 Low_i \times After(-2,-1)_t + \delta_2 Low_i \times After(0)_t + \\
& \delta_3 Low_i \times After(+1)_t + \delta_4 Low_i \times After(\geq +2)_t + \tau_1 After(-2,-1)_t + \\
& \tau_2 After(0)_t + \tau_3 After(+1)_t + \tau_4 After(\geq +2)_t + \varphi Cons_{it} + \delta_i + \gamma_t + \varepsilon_{it}
\end{aligned}
\tag{2}
$$

Among them, *After* $(-2, -1)_t$ takes 1 when the sample year is 2004 or 2003, otherwise it takes 0; *After* $(0)_t$ takes 1 when the sample year is 2005, otherwise takes 0; *After* $(+1)_t$ takes 1 when the sample year is 2006, otherwise 0; *After* $(\geq +2)_t$ take 1 when the sample year is greater than or equal to 2007, otherwise take 0, the specific results are shown in **Table 4**.

**Table 4. Empirical test results of dynamic effects.**

| Protect | Private enterprises | Local state-owned enterprises | Central state-owned enterprises |
|---|---|---|---|
| $Low_i \times$ After (-2, -1)$_t$ | -0.066 (0.042) | 0.059 (0.040) | -0.059 (0.057) |
| $Low_i \times$ After (0)$_t$ | 0.130** (0.055) | 0.127** (0.050) | 0.008 (0.070) |
| $Low_i \times$ After (+1)$_t$ | 0.123* (0.065) | 0.132*** (0.041) | 0.098 (0.071) |
| $Low_i \times$ After ($\geq$2)$_t$ | -0.086 (0.060) | 0.001 (0.001) | 0.076 (0.058) |
| Size | 0.0001 (0.0001) | 0.0004 (0.0001) | 0.040 (0.027) |
| Fixed | -0.095 (0.118) | 0.260 (0.072) | -0.0003 (0.0006) |
| Leverage | 0.001 (0.003) | 0.003 (0.004) | 0.002 (0.003) |
| Debt | -0.400*** (0.132) | -0.341** (0.055) | 0.002 (0.001) |
| Stock | 0.0001 (0.0001) | -0.004 (0.015) | 0.00003 (0.00003) |
| ROE | 0.0004** (0.0004) | 0.0003* 0.0002 | 0.002*** (0.001) |
| equity | -2.209 (3.003) | -0.002 (0.003) | -0.013 (0.010) |
| Receipt | 0.00005 (0.001) | 0.0001 (0.0001) | 0.0002* (0.0001) |
| Director | -0.520** (0.132) | -0.006 (0.068) | -0.056 (0.122) |
| Constant | 1.232 (0.046) | 1.399 (0.348) | 0.457 (0.566) |
| Firm_fe | YES | YES | YES |
| Time_fe | YES | YES | YES |
| Observations | 1642 | 3086 | 1644 |
| Rho | 0.356 | 0.262 | 0.334 |

It can be seen from the results that central state-owned enterprises are still not significant in the parallel dynamic test, and the regression coefficient of $Low_i \times After$ (-2, -1)$_t$ is not significant, that is, in the first two years of the law amendment in 2005, in 2003 and 2004, the level of creditor protection of enterprises with better profitability did not change significantly. It shows that the law amendment in 2005 passed the robustness test, and it also shows that this amendment was not predicted or reacted in advance by economic entities. The regression coefficients of $Low_i \times After$ (0)$_t$ and $Low_i \times After$ (+1)$_t$ in private enterprises and local state-owned enterprises are significantly positive, while $Low_i \times After$ ($\geq$+2)$_t$ is no longer significant, indicating that this revision does not have a lag effect of more than two years.

The dynamic test results show that the impact on the level of creditor protection is not only maintained in the year of revision, but its effect continues to extend to the year of implementation. The promotion effect is inconsistent between private enterprises and local state-owned enterprises: the dynamic effect of private enterprises in the year of revision and implementation is weakened, while the dynamic effect of local state-owned enterprises is enhanced.

This paper argues that these results are closely related to the provisions of the "piercing of the corporate veil" system. Although this amendment stipulates the "piercing of the corporate veil" system in the *Company Law*, it does not specify how to pierce and what circumstances

apply to piercing, indicating that this amendment has played a guiding and predictive role in the company's main body from the national legal policy within a certain period. The purpose is to increase the debtor's willingness to pay debts. But the dynamic timing effect is limited, under the action of external pressure, the willingness to pay debts has little effect on the promotion. The company also has a time limit on the motivation to enhance solvency, and the level of creditor protection is correspondingly time-sensitive.

The reason why the aging effect of local state-owned enterprises has been enhanced in the year of the implementation of the amendment is also consistent with the original intention of this amendment, that is, the key object of this amendment is the state-owned enterprises with more serious abuse of power. Among the state-owned enterprises, the number of local state-owned enterprises is the largest, involving a wide range of industries, and the scale of funds is more similar to that of private enterprises. In the case of limited self-owned funds, transactions with creditors are more frequent in the business process, and the sensitivity to the implementation of the company's veil system is stronger. It can be seen from the results that the level of creditor protection of local state-owned enterprises has been improved by this legal amendment, and the promotion effect of the amendment has increased over time to the year of implementation, indicating that this legal amendment has achieved the purpose of preventing state-owned enterprises from abusing corporate personality. Although local state-owned enterprises are slightly lower than private enterprises in the year of revision, the response is relatively "slow", when they realize that the regulation of the abuse of existing creditors by the amendment may lead to the effect of litigation to pursue the liability of infringing shareholders, the financial support of potential creditors will face greater competitive pressure.

Under the guidance of the decision-making of local government departments or institutions, although the response of the company's management is relatively slow, to obtain the financial support of potential creditors, there must be an incentive to reconfigure the distribution of equity and creditor's rights, which further improves the company's solvency and triggers a stronger growth effect in the future. However, central state-owned enterprises have a strong capital scale, which is much larger than that of private enterprises and local state-owned enterprises. At the same time, they are directly adjusted and supported by the central government departments, and are less sensitive to legal changes, so the results are not significant.

## Robustness tests

**Processing missing variables through Propensity Score Matching.** For enterprises, there may be many differences between enterprises with higher profitability and enterprises with lower profitability. Although these differences have been controlled in the regression model, it is still impossible to completely rule out that the significant results are caused by the differences in control variables. This paper uses Propensity Score Matching (PSM) to further address this issue. According to the existing grouping criteria, this paper uses the nearest neighbor matching method to match the similarity of covariates. The matching process adopts the 1: 1 method to construct the control group. The obtained PSM test eliminates the enterprises with large differences in covariates. After PSM matching, DID is used to test the effect of this revision. The specific results are shown in **Table 5**.

From **Table 5**, the results of the PSM-DID test modified by the *Company Law* in 2005 are significantly positive in the core explanatory variables of private enterprises and local state-owned enterprises. The results indicate that the PSM-DID test results of private enterprises and local state-owned enterprises are consistent with the results of the benchmark regression model, and the core explanatory variables of central state-owned enterprises are not significant, which proves the robustness of the above empirical test results.

**Table 5. PSM-DID test results.**

| Protect | Private enterprises | Local state-owned enterprises | Central state-owned enterprises |
|---|---|---|---|
| $Low_i \times Year2005_t$ | 0.101* | 0.103*** | 0.020 |
| | (0.057) | (0.017) | (0.014) |
| Size | 0.050 | 0.0001 | 0.0002 |
| | (0.039) | (0.0001) | (0.002) |
| Fixed | 0.082 | 0.146** | -0.0004 |
| | (0.156) | (0.064) | (0.0004) |
| Tobin | 0.002 | 0.022** | -0.006 |
| | (0.004) | (0.007) | (0.005) |
| Leverage | 0.014 | 0.004 | 0.002 |
| | (0.020) | (0.003) | (0.006) |
| Debt | 0.331*** | 0.002* | 0.015 |
| | (0.127) | (0.002) | (0.040) |
| ROE | -0.002 | 0.0001* | 0.002*** |
| | (0.0004) | 0.0002 | (0.001) |
| equity | -0.033 | -0.004 | -0.017 |
| | (0.023) | (0.003) | (0.010) |
| Receipt | -0.001 | -0.0001 | 0.001* |
| | (0.003) | (0.0001) | (0.0001) |
| Stock | 0.004 | 0.018 | 0.00004 |
| | (0.015) | (0.007) | (0.00003) |
| Director | -0.506*** | -0.065 | -0.049 |
| | (0.135) | (0.046) | (0.054) |
| Constant | 2.605 | 0.402 | 0.706*** |
| | (2.005) | (0.258) | (0.030) |
| Firm_fe | YES | YES | YES |
| Time_fe | YES | YES | YES |
| Observations | 1042 | 2157 | 1215 |
| Rho | 0.498 | 0.485 | 0.206 |

**Placebo test.** First, exclude other legal of the same rank interference. *Property Law of the People's Republic of China* came into effect on October 1, 2007. The law has more complete provisions on creditor-debtor relationships. To reduce the noise caused by policy interference, this paper excludes the sample data of the year of implementation of the *Property Law*, that is, 2007, for robustness testing. The specific results are shown in **Table 6** Panel A. The results show that excluding the impact of other relevant laws, it is still consistent with the above regression results, indicating the robustness of the results.

Second, change the grouping standard proxy variable. From the above analysis, this paper chooses the company's profitability as the grouping standard. Specifically, the return on assets (*ROA*) is used to represent the company's profitability as the grouping standard of the DID. In the robustness test, the substitution grouping standard proxy variable is the return on equity (*ROE*), and the specific results are shown in Panel B of **Table 6**. The results show that the results are still robust after changing the grouping of standard proxy variables.

Third, narrow the sample range. Some companies in the sample of this paper are listed after the starting time of the sample (2005), and the solvency of these companies may affect the impact of this inference. In the robustness test, we limit the sample to the samples listed in 2005 and before. The specific results are shown in **Table 6** Panel C. The results show that the results are still robust after narrowing the sample range.

**Table 6. Placebo test.**

| Panel A | Exclude other legal interference of the same rank | | |
|---|---|---|---|
| Explanatory variable | Private enterprises | Local state-owned enterprises | Central state-owned enterprises |
| | (1) | (2) | (3) |
| $Low_i \times Year2005_t$ | 0.133*** (0.038) | 0.240* (0.122) | -0.041 (0.205) |
| Panel B | Change the grouping standard proxy variable | | |
| Explanatory variable | Private enterprises | Local state-owned enterprises | Central state-owned enterprises |
| | (1) | (2) | (3) |
| $Low_i \times Year2005_t$ | 0.091*** (0.022) | 0.193** (0.087) | -0.026 (0.052) |
| Panel C | Narrow the sample range | | |
| Explanatory variable | Private enterprises | Local state-owned enterprises | Central state-owned enterprises |
| | (1) | (2) | (3) |
| $Low_i \times Year2005_t$ | 0.219** (0.095) | 0.248** (0.092) | -0.200 (0.501) |

## Empirical test of heterogeneous grouping

Considering that under the impact of legal changes, the level of creditor protection of enterprises with different operating conditions may have different responses, it is necessary to discuss in more depth which operating factors will further lead to the significant promotion effect of creditor protection after the 2005 revision of the *Company Law*. This paper then groups the three types of target enterprises according to different financial factors. It should be noted that this paper only lists the results of the core explanatory variables in the heterogeneity group.

**Group test on the size of debt.**    In this paper, the debt scale is grouped according to the average value of the year before the law is amended (i.e.,2004). Different types of enterprises are divided into high-debt scale groups (referred to as high-debt groups) and low-debt scale groups (referred to as low-debt groups). The grouping results are shown in **Table 7**.

As shown in **Table 7**, the core explanatory variables of the high debt group of private enterprises and local state-owned enterprises are significantly positive, which are significant at the level of 5% and 10% respectively. The core explanatory variables of the low debt group are not significant, and the results of the two groups of central state-owned enterprises are not significant. It shows that private enterprises and local state-owned enterprises with high debt scale, after the revision of the law, creditors' requirements for their ability to repay debts continue to increase.

**Table 7. Debt scale grouping.**

| Protect | Private enterprises | | Local state-owned enterprises | | Central state-owned enterprises | |
|---|---|---|---|---|---|---|
| | High-debt group | Low-debt group | High-debt group | Low-debt group | High-debt group | Low-debt group |
| $Low_i \times Year2005_t$ | 0.275** (0.118) | -0.022 (0.069) | 0.094* (0.071) | 0.066 (0.044) | 0.006 (0.079) | -0.100 (0.126) |
| Constant | 0.588 (0.069) | 1.519*** (0.086) | 0.492 (0.605) | 1.506 (0.406) | 0.446*** (0.018) | 0.360*** (0.090) |
| Firm_fe | YES | YES | YES | YES | YES | YES |
| Time_fe | YES | YES | YES | YES | YES | YES |
| Controls | YES | YES | YES | YES | YES | YES |
| Observations | 646 | 642 | 1005 | 2099 | 650 | 532 |
| Rho | 0.317 | 0.421 | 0.436 | 0.243 | 0.201 | 0.287 |

For enterprises with higher debt scale, internal financing support is limited. It is necessary to increase the scale of external financing to promote more active liquidity. It has a strong motivation to increase the level of debt financing and expand the scale of investment, which has become an important driving force after the revision of the law. Therefore, after the amendment of the law in 2005, it has significantly promoted small and medium-sized enterprises with high debt scale to improve the level of creditor protection.

**Group test on the year-on-year growth rate of operating income.** According to the average value of operating income, the heterogeneity group is divided into a high operating income growth rate group (referred to as the high-income group) and a low operating income group (referred to as the low-income group) according to the average value of operating income in the year before the law is revised (i.e.2004). The specific results are shown in **Table 8**.

As shown in **Table 8**, the grouping results show that the core explanatory variables of the high-income group in private enterprises are significant, indicating that under the external shock of the 2005 *Company Law* reform, private enterprises with higher operating income have significantly improved the level of creditor protection after the external shock of the law amendment.

This result shows that the higher the operating income of the enterprise, the better the growth ability of the enterprise. After the law is revised, the enterprise makes an effective response to the content of the law revision, which makes the profit level of the enterprise show the growth of scale effect, thus improving the solvency and promoting the protection of the interests of creditors.

However, private enterprises with a low year-on-year growth rate of operating income have limited development capacity to a certain extent, and the company's business growth trend is not obvious, resulting in the compression of the growth space of profitability. Under the influence of legal amendments, it takes a longer period to respond to legal changes at the level of the company's solvency, and the possibility of being eliminated by the market in the competition of similar enterprises increases.

Local state-owned enterprises are significant under different operating income growth rate groups, but the core explanatory variable coefficient of the high-income group is much higher than that of the low-income group. It shows that local state-owned enterprises are similar to private enterprises. After the law is amended, the high-income group can better improve the solvency of enterprises and protect the interests of creditors than the low-income group.

**Group testing of the shareholding ratio of controlling shareholders.** Considering the influence of the shareholding ratio of major shareholders on the level of creditor protection, through the heterogeneity grouping test of the shareholding ratio of the controlling

**Table 8. Business income grouping.**

| *Protect* | Private enterprises | | Local state-owned enterprises | | Central state-owned enterprises | |
|---|---|---|---|---|---|---|
| | High-income group | Low-income group | High-income group | Low-income group | High-income group | Low-income group |
| $Low_i \times Year2005_t$ | 0.222*<br>(0.118) | 0.094<br>(0.065) | 0.122*<br>(0.073) | 0.066*<br>(0.044) | 0.006<br>(0.079) | -0.100<br>(0.126) |
| *Constant* | 0.963***<br>(0.122) | 1.295***<br>(0.064) | 0.983<br>(0.773) | 1.867<br>(0.409) | 0.446<br>(0.018) | 0.360<br>(0.090) |
| *Firm_fe* | YES | YES | YES | YES | YES | YES |
| *Time_fe* | YES | YES | YES | YES | YES | YES |
| *Controls* | YES | YES | YES | YES | YES | YES |
| *Observations* | 349 | 824 | 894 | 2447 | 650 | 532 |
| *Rho* | 0.501 | 0.339 | 0.434 | 0.268 | 0.201 | 0.287 |

**Table 9. Grouping of the shareholding ratio of major shareholders.**

| Protect | Private enterprises | | Local state-owned enterprises | | Central state-owned enterprises | |
|---|---|---|---|---|---|---|
| | High proportion group | Low proportion group | High proportion group | Low proportion group | High proportion group | Low proportion group |
| $Low_i \times Year2005_t$ | 0.399*** | 0.081 | 0.144*** | 0.014 | 0.053 | 0.020 |
| | (0.125) | (0.065) | (0.056) | (0.045) | (0.079) | (0.01) |
| Constant | 1.369*** | 1.266*** | 1.630*** | 0.363*** | 0.446 | 0.105 |
| | (0.122) | (0.064) | (0.627) | (0.510) | (0.018) | (0.349) |
| Firm_fe | YES | YES | YES | YES | YES | YES |
| Time_fe | YES | YES | YES | YES | YES | YES |
| Controls | YES | YES | YES | YES | YES | YES |
| Observations | 327 | 861 | 1284 | 1773 | 676 | 520 |
| Rho | 0.598 | 0.334 | 0.434 | 0.377 | 0.301 | 0.215 |

shareholders of the enterprise, taking the year before the law revision (2004) as the standard, the shareholding ratio of the controlling shareholders is greater than or equal to 50% for the high shareholding ratio group, referred to as the high proportion group; The group with a controlling shareholder's shareholding ratio of less than 50% is the low shareholding group, which is referred to as the low shareholding group. The specific results are shown in **Table 9**.

The grouping results in **Table 9** show that the core explanatory variables of the high-proportion group of private enterprises and local state-owned enterprises are significantly positive, and the low-proportion group is not significant.

This means that in the absolute holding situation of small and medium-sized enterprises, the major shareholders have a decisive voice in the company's decision-making, and the company's property independence is more susceptible to the decision-making of the controlling shareholders. The controlling shareholder usually has the right to claim the company's remaining property. The company's profitability directly affects the company's reputation and its interests. Therefore, enterprises with relatively concentrated equity are better at making business decisions from the perspective of the company's long-term interests, are more sensitive to legal changes, and pay attention to protecting the interests of the company's existing and potential creditors.

In companies with relatively unconcentrated equity, due to the dispersion of equity, there is a more serious free-riding phenomenon among shareholders. After the law is revised, shareholders' decision-making may lag, and the sensitivity to legal changes is low, making the low-proportion group produce insignificant results.

**Group test on the market value of individual stocks.** Considering the impact of individual stock market value on the interests of creditors, the individual stock market value is divided into high stock market value group (referred to as high-value group) and low stock market value group (referred to as low-value group) according to the average value of the previous year before the legal amendment as the heterogeneity grouping standard. The specific results are shown in **Table 10**. The results show that the high-value group of private enterprises and local state-owned enterprises is significantly positive, and the low-market value group is not significant.

The stock market value represents the equity value of the company's future cash flow discounted. Small and medium-sized enterprises with high stock market value indicate that the company's expected development ability is better, the debt pressure to repay existing creditors is smaller, and the default risk is low. After the law is revised, it can quickly meet the needs of market competition, promote the improvement of the company's profitability, and thus promote the improvement of the company's solvency.

**Table 10. Group test on the market value of individual stocks.**

| Protect | Private enterprises | | Local state-owned enterprises | | Central state-owned enterprises | |
|---|---|---|---|---|---|---|
| | High-value group | Low-value group | High-value group | Low-value group | High-value group | Low-value group |
| $Low_i \times Year2005_t$ | 0.156* | -0.018 | 0.190*** | 0.001 | 0.093 | 0.023 |
| | (0.125) | (0.073) | (0.056) | (0.059) | (0.070) | (0.020) |
| Constant | 1.286*** | 1.151*** | 0.625 | 1.138 | 0.342 | 0.282 |
| | (0.093) | (0.145) | (0.496) | (1.151) | (0.068) | (0.391) |
| Firm_fe | YES | YES | YES | YES | YES | YES |
| Time_fe | YES | YES | YES | YES | YES | YES |
| Controls | YES | YES | YES | YES | YES | YES |
| Observations | 516 | 461 | 2129 | 930 | 670 | 512 |
| Rho | 0.521 | 0.592 | 0.363 | 0.467 | 0.121 | 0.281 |

The reason why the legal modification is not significant in the low market value group is that the growth of small and medium-sized enterprises with low stock market value is hindered, and the creditors and shareholders of the company have low expectations for their expected development ability, which affects the company's next investment and operation, making the legal modification of such enterprises of little significance.

## Conclusions and implications

The empirical results of this paper show that for the implementation of the system of "piercing the corporate veil", the sensitivity of different types of enterprises to legal changes is not consistent. In this paper, the 2005 *Company Law* amendment is regarded as a quasi-natural experiment. The DID is used to test the legal effect of the "piercing the corporate veil" system in this amendment. The results show that the 2005 *Company Law* amendment significantly improves the solvency of small and medium-sized listed enterprises with weak profitability and promotes the improvement of creditor protection. This legal amendment significantly improved the level of creditor protection for private enterprises with strong budget constraints and effectively solved the phenomenon of abuse of corporate personality by local state-owned enterprises. However, this amendment has no substantial impact on the protection of creditors' interests of central state-owned enterprises.

The dynamic test results further show that: the *Company Law* amendment not only played a significant role in improving the level of creditor protection of small and medium-sized enterprises in the year 2005, but also continued to play a role in improving the level of creditor protection in the year 2006, but the growth effect showed inconsistency: the growth effect of private enterprises slowed down in the year of implementation, showing the limitation of time, while the dynamic promotion effect of local state-owned enterprises was further enhanced. In addition, to further explore the impact mechanism of "piercing the corporate veil", among private enterprises and local state-owned enterprises, enterprises with higher debt scale, enterprises with a higher year-on-year growth rate of operating income, enterprises with a higher shareholding ratio of major shareholders, and enterprises with higher individual stock market value have more significantly improved the solvency of enterprises and played a better role in protecting the interests of creditors.

With the gradual improvement of the Chinese legal system, the availability of micro-enterprise data and the degree of information disclosure continue to improve. In the future, relevant research in this paper can be promoted from data of company contracts. The abuse of limited liability will not only damage the interests of creditors but also may cause damage to the interests of minority shareholders, especially the controlling shareholders who use their own

information and asset advantages to hollow out minority shareholders. In future studies, we can try to study whether the amendment of the *Company law* has a protective effect on the interests of minority shareholders, and explore the specific path to realize the protection of the *Company law* on minority shareholders.

## Author Contributions

**Conceptualization:** Jun Tian.

**Data curation:** Jun Tian, Zuopeng Chen.

**Formal analysis:** Jun Tian, Zuopeng Chen.

**Funding acquisition:** Jun Tian.

**Investigation:** Jun Tian, Zuopeng Chen, Yue Zhu.

**Methodology:** Jun Tian, Zuopeng Chen.

**Project administration:** Jun Tian, Zuopeng Chen.

**Resources:** Jun Tian, Yue Zhu.

**Software:** Jun Tian, Zuopeng Chen.

**Supervision:** Jun Tian, Yue Zhu.

**Validation:** Jun Tian, Zuopeng Chen.

**Visualization:** Jun Tian, Zuopeng Chen.

**Writing – original draft:** Jun Tian, Zuopeng Chen.

**Writing – review & editing:** Jun Tian.

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
