## [Decision Letter · Decision Letter 0]

8 Apr 2024

Piercing the Corporate Veil System and Creditors Protection: Based on the Investigation of Debt Paying Ability

——Empirical Evidence from the 2005 Company Law Amendment

PONE-D-23-43810

Dear Dr. Chen,

We’re pleased to inform you that your manuscript has been judged scientifically suitable for publication and will be formally accepted for publication once it meets all outstanding technical requirements.

Kind regards,

Gianluca Mattarocci, PhD

Academic Editor

PLOS ONE

Journal Requirements:

""Research on mechanism and countermeasures of regional employment selection of college graduates from the perspective of highly educated population loss in Heilongjiang Province"

Please respond by return e-mail so that we can amend your financial disclosure and competing interests on your behalf.

Additional Editor Comments (optional):

Reviewers' comments:

Reviewer's Responses to Questions

**Comments to the Author**

1. Is the manuscript technically sound, and do the data support the conclusions?

Reviewer #1: Yes

2. Has the statistical analysis been performed appropriately and rigorously? 

Reviewer #1: Yes

3. Have the authors made all data underlying the findings in their manuscript fully available?

Reviewer #1: Yes

4. Is the manuscript presented in an intelligible fashion and written in standard English?

Reviewer #1: Yes

5. Review Comments to the Author

Reviewer #1: I enjoyed reading the paper. I believe this research will be highly beneficial to the official authorities, companies and investors in general. I would also suggest another avenue of research on this subject to assess the piercing of corporate veil instances through other factors such as tax evasion by companies.

6. PLOS authors have the option to publish the peer review history of their article (what does this mean?). If published, this will include your full peer review and any attached files.

Reviewer #1: No

---

## [Editor Report · Acceptance letter]

26 Apr 2024

PONE-D-23-43810 

PLOS ONE

Dear Dr. Chen, 

I'm pleased to inform you that your manuscript has been deemed suitable for publication in PLOS ONE. Congratulations! Your manuscript is now being handed over to our production team.

Kind regards, 

on behalf of

Dr. Gianluca Mattarocci 

Academic Editor

PLOS ONE